# Research Progress on Neuroprotection of Insulin-like Growth Factor-1 towards Glutamate-Induced Neurotoxicity

**DOI:** 10.3390/cells11040666

**Published:** 2022-02-14

**Authors:** Lijun Ge, Shuyuan Liu, Limor Rubin, Philip Lazarovici, Wenhua Zheng

**Affiliations:** 1Center of Reproduction, Development and Aging Institute of Translation Medicine, Faculty of Health Sciences, University of Macau, Taipa, Macau 999078, China; gelijun@zcmu.edu.cn.com; 2College of Life Science, Zhejiang Chinese Medical University, Hangzhou 310053, China; liushuyuan2405@163.com; 3Allergy and Clinical Immunology Unit, Department of Medicine, Hadassah-Hebrew University Medical Center, Jerusalem 9112001, Israel; limorlaz@gmail.com; 4Pharmacology, Institute for Drug Research, School of Pharmacy, Faculty of Medicine, The Hebrew University of Jerusalem, Jerusalem 9112102, Israel; philipl@ekmd.huji.ac.il

**Keywords:** glutamate-mediated excitotoxicity, signaling pathways, insulin-like growth factor-1, neuroprotection, animal models, clinical trials

## Abstract

Insulin-like growth factor-1 (IGF-1) and its binding proteins and receptors are widely expressed in the central nervous system (CNS), proposing IGF-1-induced neurotrophic actions in normal growth, development, and maintenance. However, while there is convincing evidence that the IGF-1 system has specific endocrine roles in the CNS, the concept is emerging that IGF-I might be also important in disorders such as ischemic stroke, brain trauma, Alzheimer’s disease, epilepsy, etc., by inducing neuroprotective effects towards glutamate-mediated excitotoxic signaling pathways. Research in rodent models has demonstrated rescue of pathophysiological and behavioral abnormalities when IGF-1 was administered by different routes, and several clinical studies have shown safety and promise of efficacy in neurological disorders of the CNS. Focusing on the relationship between IGF-1-induced neuroprotection and glutamate-induced excitatory neurotoxicity, this review addresses the research progress in the field, intending to provide a rationale for using IGF-I clinically to confer neuroprotective therapy towards neurological diseases with glutamate excitotoxicity as a common pathological pathway.

## 1. Introduction

Insulin-like growth factor 1 (IGF-1) is considered an attractive therapeutic alternative for diverse neurological pathologies since it plays a key trophic role in the developing nervous system and maintenance and regulation of neurological functions in the adult brain [1]. It is also recognized to date that IGF-1, together with other neurotrophic growth factors, plays the first line of defense in the adult brain exposed to neurotoxic insults. Abnormalities in IGF-1 levels have been implicated in a variety of neurological and psychiatric disorders. We review here selected literature suggesting that IGF-1 plays a unique role in neuroprotection towards glutamate-induced excitotoxicity in in vitro and in vivo models. Specifically, we focus on the relationship between IGF-1-induced neuroprotection and glutamate-induced excitatory neurotoxicity, present the reciprocal cross-talk between IGF-1 and glutamate receptors, and briefly address preclinical and clinical studies providing pieces of evidence that IGF-I confers neuroprotection in animal models and neurological diseases [2]. Excitotoxicity is a phenomenon that describes the toxic actions of excitatory neurotransmitters, primarily glutamate, where the exacerbated or prolonged activation of glutamate receptors starts a cascade of neurotoxicity that ultimately leads to neuronal cell death and the loss of neuronal function. In this process, the shift between normal physiological function and excitotoxicity is largely controlled by astrocytes since they regulate the levels of synaptic glutamate. The molecular mechanism that triggers excitotoxicity involves alterations in glutamate and calcium metabolism and dysfunction of glutamate receptors. Excitotoxicity is the cause but also the consequence of other cellular pathophysiological processes, such as mitochondrial dysfunction, neuronal damage, and oxidative stress. It is known that the excessive activation of glutamate receptors results in the sustained influx of calcium into neurons that leads to several deleterious consequences, including mitochondrial dysfunction, overproduction of reactive oxygen species (ROS), the impairment of calcium buffering, and the release of pro-apoptotic factors, among others, that cumulatively contribute to neuronal loss. Recent studies implicate glutamate-induced excitotoxicity as a central mechanism in the pathogenesis of many neurodegenerative diseases, including amyotrophic lateral sclerosis, Alzheimer’s disease, traumatic brain injury, and epilepsy, suggesting that neurodegenerative diseases may share excitotoxicity as a common pathological mechanism [3]. Thus, IGF-1 signaling involved in neuroprotection towards glutamate-induced excitotoxicity is also of critical significance for the future clinical treatment of many neurodegenerative diseases [4]. Considering that alterations of IGF-1 levels have been implicated in human ischemic stroke [5] and brain trauma [6], and exogenous administration of IGF-1 has neuroprotective effects in animal models of ischemia [5], the investigations of the relationship between IGF-1 levels and activity and neuronal injury is of great value. They may clarify the neuroprotective role of IGF-1 [7] in glutamate-induced excitatory toxicity and allow the implementation of the findings towards novel neuroprotective therapeutic strategies, with the understanding that the targets are not specific symptoms, but the underlying molecular signaling pathways and cellular phenomena of excitotoxicity.

## 2. IGF-1 and IGF-1R

IGF-1 and IGF-2 were discovered in 1957 by Salmon and Daughaday [8] and named “sulphation factors” by their property to stimulate sulphate incorporation into the rat cartilage. Froesch et al. characterized the non-suppressible insulin-like activity (NSILA) of two soluble serum components [9]. In 1972, the definitions “sulphation factor” and “NSILA” were replaced by the term “somatomedin”, defining a hormone under the control, and mediating the effects, of growth hormone (GH) [10]. In 1976, Rinderknecht and Humbel [11] isolated two active substances from human serum, with the N-terminal amino acid sequences of Gly-Pro-Glu- in NSILA I, and Ala-Tyr-Arg- and Tyr-Arg- in NSILA II, that, due to their structural resemblance to proinsulin, were coined “insulin-like growth factor 1 and 2” (IGF-1 and 2). Insulin-like growth factors (IGFs) not only promote cell differentiation and proliferation but also have insulin-like effects. As hormones they exert systemic influence and as autocrine/paracrine factors they exert local effects [12]. The IGF hormonal system includes IGF-1, IGF-2, and their corresponding receptors (IGF-1R, IGF-2R), as well as six insulin growth factor binding proteins (IGFBPs) [13,14]. IGF-1 is a single-chain polypeptide composed of 70 amino acids, connected by three pairs of disulfide bonds, and a molecular weight of 7.6 kDa, and are composed of three helical segments which are connected by a 12-residue linker, known as the C-region [15] (Figure 1). To date, human recombinant IGF-1 (rhIGF-1; Mecasermin, Increlex) for clinical use [16] is efficiently produced and formulated in *Escherichia coli* [17]. IGF-1, IGF-2, and insulin exert their activities by binding to different, but highly homologous (~75%), ~450 kDa (αβ)2 dimeric, tyrosine-kinase IGF-1 receptors (IGF-1R; IGF-2R), and insulin receptors (IR-A; IR-B). An IGF1 receptor is characterized by a high affinity for IGF-1 (IC_50_ of 0.2–0.8 nM) and IGF-2 (IC_50_ of 0.5–4.4 nM), but it can also bind insulin with 50- to 100-fold lower affinity (IC_50_ over 30 nM). Insulin/IGF signaling cross-talk is also amplified by the heterodimerization of IGF-1R and IR-A, and the presence of hybrid receptors that can be effectively activated by IGF-1, but not insulin [18]. It is important to stress that the availability of free IGF-1 and IGF-2 for receptor signaling is modulated by IGFBPs. Under normal physiological conditions, less than 1% of IGF-1s exist in a free form, and the vast majority of IGF-1 is bound to IGFBPs [19]. This results in prolongation of the half-life of IGF-1 in serum and the ability of IGF-1 to cross the blood-brain barrier by a transport system that functions in synchrony with IGFBP, in the periphery of the nervous system, to regulate the availability of IGF-1 in the CNS [20].

The vast majority of IGF-1 is produced in the liver and is regulated by growth hormone (GH), secreted into the blood by the pituitary glands, responsible for a negative feedback transcription regulation of GH, via POU1F1/CREB binding protein interactions [21]. The CNS receives both endocrine and paracrine inputs from IGF-1. All major CNS cell types, particularly in the cortex, hippocampus, cerebellum, hypothalamus, subventricular zone, and dentate gyrus, produce IGF-1 [22,23,24]. IGF-1, as an important neurotrophic factor, is involved in regulating neuronal growth, development, metabolism, regeneration, and neuroplasticity, a term describing the adaptive changes made by the CNS in the face of changing functional demands in processes such as learning and memory [22,25].

IGF-1R is a receptor-type tyrosine kinase, composed of α and β membrane-spanning glycoproteins [26] that bind the IGF-1 ligand. After ligand occupancy, the receptor undergoes dimerization, which in turn activates the tyrosine kinase’s phosphorylation activity of the downstream substrates involved in the intracellular transmission of IGF-1 signals [27]. Activated IGF-1R phosphorylates several substrates that lead to the binding and activation of downstream signaling pathways, PI3K/Akt/mTOR pathway, and Ras/Raf/MAPK pathway [28], required for the induction of various bioactivities of IGFs, including cell proliferation, cell differentiation, and cell survival (Figure 2).

## 3. Glutamate Excitotoxicity

Glutamate is the main excitatory neurotransmitter in the brain, with several types of receptors found throughout the central nervous system having important roles in memory, cognition, mood regulation, and motor activity [29]. This crucial excitatory amino acid is extensively recycled between neurons and astrocytes in a process known as the glutamate-glutamine cycle. This cycle is an open cycle, meaning that intermediates are lost, particularly due to a substantial oxidative metabolism of glutamate in both neurons and astrocytes [30]. Under normal physiological conditions, glutamate synaptic concentration is a necessary factor for maintaining excitatory basal and stimulated neurotransmission. In many polysynaptic neuronal pathways, the electrical impulses are depolarizing the glutamatergic neurons, resulting in the release to the synaptic cleft of the glutamate stored in the vesicles [31]. Thereafter, glutamate binds and activates postsynaptic membrane receptors followed by signal termination due to glutamate uptake by astroglia. Glutamate in astrocytes is converted to glutamine by glutamine-synthetase. The generated glutamine is transferred from astrocytes to neurons through a glutamine transporter. Once glutamine enters the neuron, it is converted to glutamate by glutaminase in the mitochondria. Finally, cytosolic glutamate is concentrated in synaptic vesicles through vesicular glutamate transporters, thus completing the glutamate-glutamine cycle [32]. Astrocytes maintain the glutamate homeostasis in the CNS by controlling the fine balance between glutamate uptake and release [33] (Figure 3).

Under physiological conditions, glutamate produces an excitatory, electrophysiological response by differential binding of the different presynaptic and postsynaptic receptors. Excessive, persistent activation of glutamate receptors results in neuronal dysfunction and cell death, a process called excitotoxicity, which involves calcium overload, mitochondrial damage, and oxidative stress [34]. Although the mechanism of excitotoxic injury is not fully understood, it has been proposed that one mechanism involves chronic dysregulation of the glutamate-glutamine shuttle, which plays a key role in excitotoxicity-induced cell death [35]. The excitatory effects of glutamate are induced by the activation of three major types of ionotropic receptors and several classes of metabotropic receptors linked to G-proteins [36]. In addition, it was further proposed that glutamate-induced excitotoxicity is mainly caused by excessive Ca^2+^ influx mediated by NMDARs, which are thought to be more permeable to Ca^2+^ than other ionotropic glutamate receptors [37].

### 3.1. NMDAR Mediated Excitotoxicity

N-methyl-D-aspartate receptor (NMDAR) is an ionotropic glutamate receptor, which is a tetramer composed of a NR1 subunit, a NR3 subunit bound to glycine, and a NR2 bound to glutamate [38]. NR1 is the basic subunit of ion channels, NR2 is the regulatory subunit, and the NMDA receptors composed of different NR2 exhibit different brain distribution and physiological characteristics. This receptor is a calcium channel, which is located in synaptic and extra-synaptic sites, triggering different signaling cascades. The degree of excitotoxicity depends on the magnitude and duration of synaptic and extra-synaptic NMDAR co-activation [39]. Under resting conditions, the channel pores of NMDARs are blocked by Mg^2+^. Upon glutamate release from presynaptic sites, activated receptors cause a partial depolarization in the postsynaptic membrane that is sufficient to remove the Mg^2+^ block from NMDARs, enabling an influx of Na^+^ and Ca^2+^ into the neuron. The Ca^2+^ influx through NMDARs is not only critical for the normal physiological processes in neurons but also plays a major role in initiating neurotoxicity [40]. In excitotoxicity, excess glutamate release results in the over-activation of NMDARs that leads to the calcium overload inside the neurons. Calcium overload triggers, in turn, a range of downstream neuronal cell death signaling events such as calpain activation [41], reactive oxygen species (ROS) generation [42], and mitochondrial dysfunction [43], resulting in neuronal aponecrosis. The calcium entry through extra-synaptic NMDARs contributes to calcium overload in the mitochondria. The mitochondria, besides their role in ATP production, participate in calcium homeostasis, acting as a buffering organelle. Disruption of mitochondrial calcium homeostasis has either been linked to neuronal cell death by triggering apoptosis, or driven by the opening of the mitochondrial transition pore [44]. Activation of extra-synaptic NMDAR depends on multiple conditions, such as their topographic location and activity in neurons, the activity of transporters in glial cells, and the overflow of glutamate at synaptic sites. Interestingly, neuronal gap junctions [45] and clathrin-dependent endocytosis [46] participate in some types of NMDA receptor-mediated excitotoxicity. To date, the hypothesis relevant for glutamate-induced excitotoxicity claims a dual role of NMDARs in cell survival and death. Activation of NMDARs can trigger survival or cell death signals, depending on the subcellular locations or subtypes of NMDARs [47]. In mature neurons, NR2A-containing NMDARs are abundant in the synapses, and NR2B-containing NMDARs are enriched in the extrasynaptic sites. In general, synaptic/NR2A-containing NMDARs are associated with pro-survival effects, whereas extrasynaptic/NR2B-containing NMDARs are involved in pro-death signaling [48].

### 3.2. AMPAR Mediated Excitotoxicity

The α-amino-3-hydroxyl-5-methylisoxazole-4-propionic acid subtype ionotropic glutamate receptors, (AMPA receptors), are composed of four subunits, GluA1 to GluA4 (NRA1-4). These NRA1/3/4 subunits, but not NRA2, have high permeability for Ca^2+^. The majority of AMPA receptors in vivo contain GluR2 subunits whose ion selectivity is dominant over other subunits [49,50]. AMPAR mediates fast synaptic transmission at excitatory synapses, while NMDAR is critical in producing several different forms of synaptic plasticity [51]. Upregulation of calcium-permeable AMPARs, together with their biological amplifying effects, triggers the Ca^2+^ influx overload and excitotoxicity, which can lead to mitochondrial injury, endoplasmic reticulum (ER) stress, activation of apoptotic cascades, and cell death [52]. Recent studies have found that compared with other neurons, motor neurons are more vulnerable to AMPAR-mediated neurotoxicity [53]. For example, adult mice develop posterior limb paralysis and bilateral motor neuron degeneration within a few days after continuous injection of AMPA in the lumbar spinal cord, which may be due to the low buffering capacity of motor neuron Ca^2+^ and the absence of NRA2-lacking AMPARs [54]. By contrast, GluA2-lacking AMPARs at mPFC synapses may also mediate altered outcome predictions after cocaine self-administration (SA) [55]. Similarly, in D1-MSN, the presence of GluA2-lacking AMPARs were observed after mice exposed to various regimens of cocaine. By increasing excitatory transmission onto D1R-expressing MSN, an imbalance between direct and indirect pathway output from the NAc may be created. In addition, the insertion of GluA2-lacking AMPAR not only strengthens the inputs but also makes them calcium permeable [56]. Interestingly, recent studies have found that AMPARs with NRA1 were very important to induce long-term potentiation (LTP), which was the main target of calcium/calmodulin-dependent kinases II [57,58]. Calcium-permeable AMPAR-associated pathological alterations could induce neural excitotoxicity in different brain regions, neural circuits, and cellular types, as well as various intracellular signaling pathways, all of which may correspondingly lead to some unique manifestations of neurological diseases [59].

### 3.3. Calcium Channels Mediated Excitotoxicity

During glutamate-induced excitotoxicity, elevated levels of extracellular glutamic acid cause persistent depolarization of the neuron. This triggers a cascade of parallel cellular events that lead to cell death. Several key mechanisms of this cascade have been identified: events depending on sodium influx, events depending on exocytosis of glutamate, and events depending on calcium influx. It is plausible that sodium entry is responsible for early necrotic cell death events, calcium entry for delayed apoptotic events, and exocytosis of glutamate followed by activation of glutamate receptors, for synergic-amplification of these aponecrotic, neurodegenerative processes. Since the first descriptions of glutamate-induced excitotoxicity, it has been clear that the key intracellular driver of these mechanisms is a massive increase in the intracellular calcium concentration [60]. In addition to the NMDARs and calcium-permeable AMPARs responsible for the massive calcium influx, voltage-dependent calcium channels (VGCC, Cav), activated by depolarization, are also involved in glutamate-induced excitotoxicity [61]. However, calcium influx via VGCCs is much less toxic than influx via NMDARs [62]. VGCCs are classified into two several groups: the CaV1 subfamily (CaV1.1 to CaV1.4) including channels that mediate L-type Ca^2+^ currents; the CaV2 subfamily (CaV2.1 to CaV2.3) including channels that mediate P/Q-type, N-type, and R-type Ca^2+^ currents, respectively; and the CaV3 subfamily (CaV3.1 to CaV3.3) including channels that mediate T-type Ca^2+^ currents [63]. All, except the CaV3 (T type) channels, are associated with several auxiliary subunits, termed α 2δ, that have been found to interact with the NMDA receptors NR2A and NR2B subunits [64]. In brain slices of mouse somatosensory cortex, it has been found that a Ca^2+^ influx through CaV 2.1 (P/Q-type) channels is directly correlated with glutamate release and activation of NMDA receptors, a process responsible for K^+^ depolarization-induced cortical spreading depression [65]. In addition, Neuronal protection against glutamate excitotoxicity was also found following treatment of primary corticostriatal neurons in mouse brains with Cav1 blockers [66]. In a small subset of cortical and hippocampal neurons, characterized by elevated expression of VGCCs and enhanced voltage-gated calcium currents, and mitochondrial dysfunction, depolarization evoked stronger calcium elevations, approaching those induced by NMDA [62]. These studies exemplify some of the contributions of calcium channels to glutamate-mediated excitotoxicity.

## 4. The Reciprocal Cross-Talk between IGF-1R and Glutamate Receptors

Glutamate-induced excitotoxicity during brain injury in ischemic stroke and trauma leads to a significant increase in expression of IGF-1 in astrocytes [67], microglia [68,69], and neurons [70], followed by a significant stimulation of IGF-1R phosphorylation [71]. IGF-I autocrine and paracrine functions are mediated by activation of IGF-IR, which in turn leads to the activation of downstream signaling pathways, PI3K-kinase pathway, and Ras–MAPK pathway. Exogenously applied IGF-1 was found to be neuroprotective, reducing neuronal loss and improving both motor and cognitive neurological outcomes, in animal models of hypoxic-ischemic and traumatic brain injuries [72,73,74]. These findings led to the proposal that IGF-1, or its agonists, may be used as therapeutics to improve outcomes following brain injury in brain stroke, trauma, and other neurodegenerative diseases [25,75,76]. However, the mechanistic molecular and cellular gaps in understanding IGF-1-induced neuroprotection require investigations on IGF-1R signaling cross-talk with glutamatergic receptors. This understanding may speed up the progress in clinical translation of IGF-1 for developing new neuroprotective therapies towards neurological diseases with glutamate excitotoxicity as a common pathological pathway.

In NMDA-induced excitotoxicity (autophagy cell death) of cultured hippocampal neurons, IGF-1 pre-treatment conferred neuroprotection, dependent on the PI3K-AKT-mTOR signaling pathway [77,78,79]. Considering that: i. the cellular effects of IGF-I are mediated by the insulin receptor substrate (IRS) proteins [80]; ii. IRS2 deficiency impairs activation of the NR2B subunit of NMDA receptors; and iii. Akt phosphorylates NR2C, and, unlike NR2A and NR2B, supports neuronal survival [81], it is plausible to hypothesize that IGF-1-induced neuroprotection towards NMDA-induced excitotoxicity, is mediated by IRS2 [82], and by the site-specific phosphorylation of NR2B and/or NR2C. Since NMDA-induced excitotoxicity is also mediated by the stimulation of the tyrosine kinase Fyn–NR2B–CaMKII pathway, and IGF-1 suppressed this pathway by phosphorylating Ser1303 of NR2B [83], a negative feedback mechanism was proposed between IGF-1R and the different phosphorylation sites of extrasynaptic NR2B, thus explaining the neuroprotection of IGF-1 towards NMDA-induced excitotoxicity (Figure 4).

Complementary studies indicated physiological antagonistic effects of glutamate-activated NMDARs on the IGF-1 receptors. During excitotoxicity, the excessive glutamate, by binding and activation of NMDAR-NR2B, decreased the phosphorylation of tyrosine residues 1131, 1135/1136, 1250/1251, and 1316, while it did not affect tyrosine 950 in the cortical neurons’ IGF-1R [84]. This finding is indicative that glutamate may have various effects on different phosphorylation sites of IGF-1 receptors that may impact IGF-1 signaling, since phosphorylation sites of IGF-1R are linked to its various anti-apoptotic, neuroprotective effects [85]. In line with these findings, studies in mice have shown that when glutamate was injected intracerebroventricularly, decreased phosphorylation of IGF-1 receptors and Akt resulted, and that this effect was reversed by the NMDA antagonist MK-801, but not by the non-NMDA antagonist. Similarly, an endogenous glutamate increase that was induced by focal cerebral ischemia, gradually reduced the phosphorylation of IGF-1 receptors and Akt, shortening the therapeutic window of IGF-1 [86]. In another approach, it was found that oxidative stress (H_2_O_2_ treatment), as occurring during glutamate excitotoxicity, attenuated IGF-1R tyrosine phosphorylation and its survival signaling properties via NMDAR- NR2B receptors in SH-SY5Y human neuroblastoma and primary-cultured cortical neurons. In this study, small interference RNA (siRNA) for NR2B, blocked the effect of glutamate on IGF-1R phosphorylation, while the NR2As siRNA treatment was not effective [87]. These findings are consistent with our early findings using rat hippocampal neuronal cultures, in which glutamate, by acting on NMDAR, attenuated IGF-1 receptor tyrosine phosphorylation and its survival signaling properties [88]. Glutamate-induced uncoupling of IGF-I signaling, by phosphorylating the IGF-I receptor docking protein insulin-receptor-substrate (IRS)-1 on Ser 307, through a pathway involving activation of PKA and PKC, may constitute an additional route contributing to excitotoxicity [89]. Interestingly, it was also reported that metabotropic glutamate receptor 2 (mGlu2) may cause transactivation of IGF-1R through phosphorylation by the FAK-tyrosine kinase [90]. These selected studies exemplify the existence of a feedback mechanism between the extrasynaptic NR2B and IGF-1R, counteracting the neuroprotection of IGF-1 towards NMDA-induced excitotoxicity (Figure 4). This concept is further supported by the “loss of function” studies. Astrocytes regulate many aspects of the brain microenvironment, including controlling glutamate-glutamine cycling, which ultimately supports neuronal metabolism, neurotransmission, and neuroprotection from excitotoxicity. Using small-molecule, IGF-1R inhibitors, and Cre-driven genetic approaches to reduce IGF-1R in vivo and in vitro in cultured rodent astrocytes, a significant reduction in the glutamate uptake was observed, due to a significant decrease in the expression levels of the glutamate transporter. These data indicate that reduced IGF-1 signaling will favor an accumulation of extra-synaptic glutamate, which may contribute to glutamate-excitotoxicity in disease states where IGF-1 levels are low [91].

## 5. IGF-1 Modulation of Glutamate-Induced Synaptic Plasticity

Synaptic plasticity involves both short-term and long-term processes. The short-term synaptic plasticity includes facilitation, depression, and potentiation, and the long-term synaptic plasticity includes long-term potentiation (LTP) and long-term depression (LTD) [92], which can lead to synaptic dysfunction, causing learning and memory impairment once the two processes are unbalanced [93]. NMDARs and AMPARs are important excitatory receptors for synaptic transmission and plasticity. IGF-1 is one of the neurotrophic factors that is maintaining glutamatergic synaptic function by stimulating the PI3K/Akt or MAPK/Erk signaling pathway [94]. Moreover, IGF-1 is inducing phosphorylation of glycogen synthase kinase 3 beta (GSK3β) at serine-9 and thus causing its inactivation, a critical convergence event in the promotion of survival of the glutamatergic, cerebellar granule brain neurons [95]. Since GSK3β mediates the interaction between the two major forms of synaptic plasticity in the brain, NMDAR-dependent long-term potentiation (LTP) and NMDA receptor-dependent long-term depression (LTD) [96], the IGF-1 induced stimulation of the phosphorylation of GSK3β significantly affects the synaptic plasticity. Moreover, the synaptic plasticity mechanisms of IGF-1 may be accomplished by the modulation of the brain levels of the brain-derived neurotrophic factor [97], the increase in calcium influx through L-type calcium channels, and the activation of CaMKIIα [98,99], as well as by the decrease in the GABA-A receptor- alpha-1 subunit expression [100], the regulation of the astrocytes’ glutamate-transporters [93], and the cooperative interactions with different neurotrophins [101].

AMPAR is expressed in a wide range of brain glial cells, besides neurons, where they regulate important cellular functions. AMPAR allows glial cells to sense the activity of neighboring neurons and synapses, rendering the glial cells sensitive to elevations of the extracellular concentration of glutamate, thus triggering neuronal pathophysiological responses and amplifying neuronal excitotoxicity [102]. The glutamate concentration and cellular localization of AMPAR along with IGF-1, 2 expressions were upregulated in the periventricular white matter (PWM) of neonatal rats exposed to hypoxia injury [103]. In primary microglial cultures subjected to hypoxia in vitro, administration of exogenous glutamate decreased IGF-1, suggesting that increased IGF-1expression may represent an early protective mechanism in attenuating the hypoxic damage, but a subsequent glutamate-induced decrease of IGF-1 expression may cause cell death due to excitotoxicity [102]. Intraperitoneal injections of IGF-1, over two weeks, reversed deficits in hippocampal AMPA signaling, LTP, and motor performance in Shank3-deficient mice [104].

Shank3 is part of the glutamate receptor body, which physically connects the parent-NMDA receptor to the metabolite mGlu5 receptor by interacting with the scaffold-folding protein PSD95-GKAP-Shank3-Homer [105]. These findings may suggest an important role of IGF-1 on correcting the integrity of the glutamate receptosome required for the synaptic transmission and plasticity.

## 6. IGF-1 Modulates Calcium Pathway

IGF-1 induces within seconds a large, tyrosine-, kinase-dependent increase in calcium channel currents in cerebellar granule neurons. While P, Q, and R channels were unaffected, N and L channel activities were significantly potentiated at specific membrane voltages. Moreover, transient expression of the dominant negative and wild-type phosphatidylinositol 3-OH kinase (PI3K) subunits, as well as the application of specific inhibitors, suggest that the role of PI3K on IGF-1 is critical, indicating that the regulation of N and L calcium channels may control calcium-dependent processes, such as neurotransmitter release and IGF-1-dependent survival [106]. In the cortex and hippocampal neurons, depolarization and IGF-1 rapidly increase phosphorylated-CREB levels, which require CaV1.3 activity and the S1486 phosphorylation site to achieve a full effect [107]. In addition, IGF-1 promotes the survival of cerebellar granule neurons by enhancing calcium influx through L-type calcium channels increased CaMK-IV activity, which acts to decrease nuclear transcription factor CCAAT enhancer-binding proteins (C/EBPβ). Conversely, NMDA receptor-mediated influx rapidly elevates nuclear C/EBPβ and induces excitotoxic death via activation of the calcium-dependent phosphatase, calcineurin (Figure 5). Moderate levels of AMPA receptor activity stimulated L channels to improve survival, whereas higher levels stimulated NMDA receptors and reduced neuronal survival, suggesting differential synaptic effects. Finally, N-type calcium channel activity reduced survival, potentially by increasing glutamate release [97]. The Na^+^/Ca^2+^ exchanger (NCX) is an important bidirectional transporter of calcium in neurons and is involved in neuroprotection. In rat primary neuronal cultures, IGF-1 produced an increase in the NCX-mediated inward current and a decrease in the NCX-mediated outward current, indicative of its involvement in IGF-1-induced neuroprotection [108]. Therefore, the neuroprotective effects of IGF-1 on neurons can be achieved by regulation of several subtypes of calcium channels, which in turn modulate the expression and activity of CaMKs and of specific nuclear transcription factors regulating genes involved in neuronal calcium homeostasis, and maintaining the survival of neuronal cells.

## 7. IGF-I Confers Neuroprotection towards Neurological Diseases with Glutamate Excitotoxicity

IGF-I exerts its pleiotropic neuroprotective functions in an endocrine, autocrine, and paracrine fashion [109]. Numerous research studies indicated that the reduced IGF-1 levels in the serum followed by decreased activity of IGF-1 signaling pathways plays a significant role in the progression of many neurological disorders, including those with glutamate excitotoxicity as a common pathological pathway such as ischemic stroke and traumatic brain injury [110]. Clinically, several studies have shown that reduced levels of IGF-1 in human patients correlated with increased mortality rate, poorer functional outcomes, and increased morbidities following an ischemic stroke [5]. In animal models of ischemia, administering exogenous IGF-1 using various routes of administration (intranasal, intravenous, subcutaneous, or topical) at various time points before and/or following the insult, attenuated the neurological damage and accompanying behavioral changes (Table 1) [104,111,112,113,114,115,116,117,118,119,120,121,122,123,124,125,126,127]. Therefore, since dysregulation of IGF-1 signaling was a common observation in neurodegenerative manifestations of excitotoxicity, the clinical rationale proposed that restoration of abnormal IGF-1signaling by exogenous supplementation could result in neuroprotection and neurotrophic effects for many clinical-pathological presentations. For this purpose, different delivery routes and therapeutic protocols were used in clinical trials on human patients treated with human recombinant IGF-1 or its analog Trofinetide [glycyl-L-methylprolyl-L-glutamic acid (NNZ-2566)] [128,129]. Multiple studies show mixed pieces of evidence with regards to serum IGF-1 concentration and the long-term neuroprotective effects, tolerability, safety, and efficacy of IGF-1 in many CNS disorders, most notably stroke, traumatic brain injury, amyotrophic lateral sclerosis, Alzheimer’s disease, autism spectrum disorder, and others. Table 1 presents preclinical and human clinical studies and trials providing strong evidence that IGF-I confers neuroprotection in preclinical experimental animal models and clinical trials in human patients with different neurological diseases. However, there are many reservations about these clinical evaluations. Serum IGF-1 may not adequately reflect the concentration of IGF-1 within the brain and there are methodological variations between studies measuring IGF-1 itself, with some measuring total level and others just the free amount [130]. Interestingly, one prospective population-based study found no direct association between IGF-1 and cognition over 20 years in 746 men [131]. Much larger, prospective longitudinal clinical studies are required to establish not just correlation, but determine any direction of causation, if it exists, between IGF-1 treatment and a clinical neuroprotective, therapeutic effect.

## 8. Summary

Excess activation of ionotropic glutamate receptors, followed by calcium overload and oxidative stress due to accumulation of reactive oxygen and nitrogen species, which further leads to mitochondrial dysfunction, lipid peroxidation, and oxidation of proteins and DNA, triggers neuronal aponecrotic cell death in the central nervous system under physiological and pathological conditions. Evidence is emerging that neuronal loss following glutamate-induced excitatory neurotoxicity propagates through distinctive, and mutually exclusive signal transduction pathways. IGF system components are widely expressed in the nervous system where there is substantial evidence for neuroprotective and neurotrophic actions of IGF-I. Some of the signaling pathways beneficially modulated by IGF-1 to confer neuroprotection include PI3K/Akt/mTOR [77], Ras/Erk1/2 [132], GSK3B/NF-kB/NLRP3 [133,134] and L-type calcium channel/CaMK II and IV that activate CREB [135]and C/EBPβ [98]. Moreover, IGF-1 can regulate the glutamate-glutamine cycle and ameliorate glutamate-induced excitotoxicity, but glutamate can also increase damage to nerve cells by reducing the neuroprotective effects of IGF-1. These regulatory reciprocal mechanisms include i. IGF-1 improvement of the LTP effect by GSK3β phosphorylation, blocking NMDAR dependent-LTD; ii. IGF-1 enhancement of AMPAR dependent-LTP, by activating the phosphorylation of AMPARs; iii. IGF-1 modulation of several subtypes of calcium ion channels to regulate calcium homeostasis; and iv. glutamate activation of NMDAR- NR2B, reducing the phosphorylation levels of IGF-1R, weakening the survival-promoting, neuroprotective effect of IGF-1; v. NMDAR-induced serine phosphorylation of IRS, dysregulating IGF-1 signaling and preventing its neuroprotective effects. IGF-1 ameliorates neuronal oxidative stress by increasing the synthesis of the antioxidant glutathione [136] and increases the expression of neurotrophins’ trk receptors [137] towards amplification of the neuroprotective effect. Moreover, IGF-1 increases the proliferation of the brain’s neuronal progenitors [138]and increases hippocampal neurogenesis [139]. This property, and the ability of IGF-1 to stimulate neurite outgrowth [140], may contribute to neuronal regeneration [141] after glutamate-induced excitotoxicity. Animal and clinical trials have proven the safety, tolerability, and efficacy of IGF-1, and the exploration of the cross-talk pathways between IGF-1 and glutamate are ongoing, suggesting that IGF-1 may become a potential drug for the treatment of neurological diseases with glutamate excitotoxicity as a common pathological pathway (Figure 4 and Figure 5). Despite the progress in the therapeutic use of IGF-I, the mechanisms of IGF-I-induced neuroprotection are not yet fully elucidated and additional clinical trials using larger cohorts of human patients need to be conducted. Insulin-like Growth Factor-1 (IGF-1) is neuroprotective and improves long-term function after glutamate-induced excitotoxicity. However, from a pharmaceutical point of view, its clinical application for the therapy of neurological disorders is disadvantageous due to its large molecular size, poor CNS uptake, and mitogenic potential. Small, IGF-1 mimetic, cyclic peptides have advantages over IGF-1, representing a novel strategy of pharmaceutical discovery for neurological disorders [142]. In conclusion, it is expected that IGF-1 and derived peptidomimetics with pleiotropic effects will provide a new, supplemental therapy that will target specific excitotoxic processes and brain cell types that contribute to different neurological diseases.

## Figures and Tables

**Figure 1 cells-11-00666-f001:**
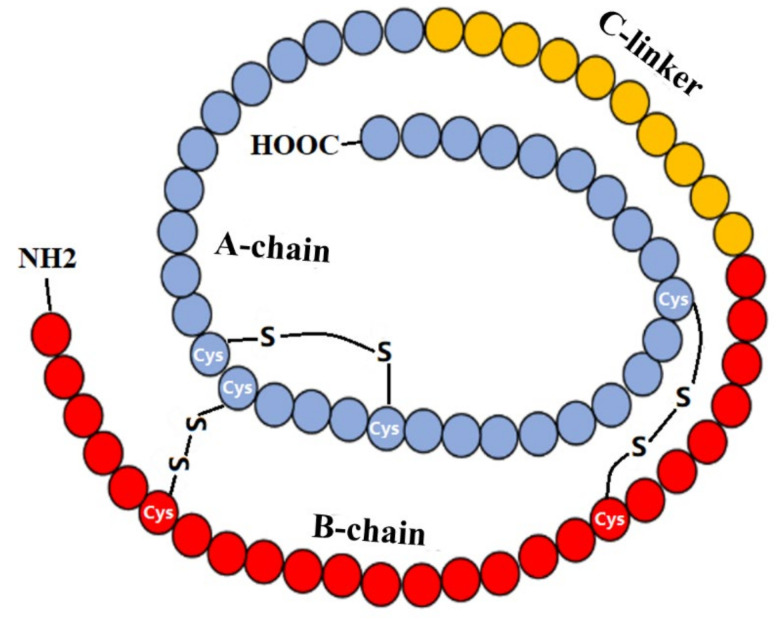
Scheme of IGF-1 Structure. IGF-1 is a single peptide chain, composed of 70 amino acids; three pairs of disulfide bonds connect A- and B-chains, in which the A-chain of 35 amino acids (blue) contains the carboxyl-terminal and the B-chain of 25 amino acids (red) contains the amino-terminal. The 12-residue linker, known as the C-linker (yellow), connects A- and B- chains.

**Figure 2 cells-11-00666-f002:**
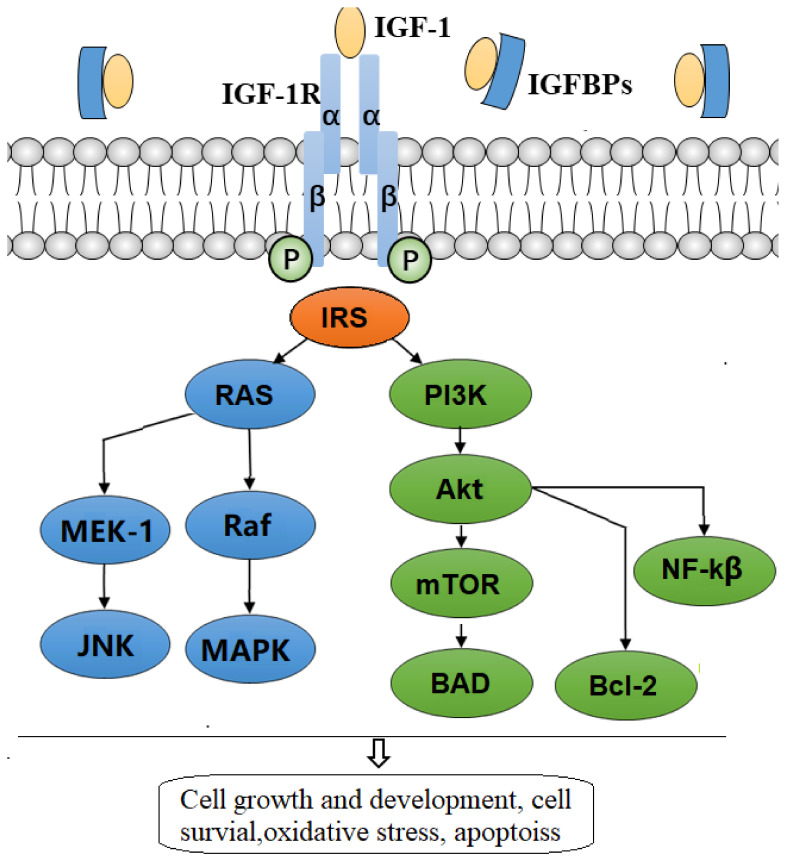
Schematic diagram of commonly accepted IGF-signaling. The half-life and other biological activities of Insulin-like growth factor-1 (IGF-1) is regulated by binding to insulin growth factor binding protein (IGFBP), while free IGF-1 can bind specifically to the IGF1 receptor. Ligand binding to the α-subunit of the receptor leads to a conformational change in the β subunit, resulting in the activation of the receptor’s tyrosine kinase activity. Activated receptor phosphorylates several substrates, including insulin receptor substrates (IRSs) and Src homology and collagen family protein (SHC). Phospho-tyrosine residues in these substrates are recognized by certain Src homology 2 (SH2) domain-containing signaling molecules. These include phosphatidylinositol 3-kinase (PI 3-kinase), growth factor receptor-bound 2 (GRB2), and others. These bindings lead to the activation of downstream signaling pathways, PI3K/AKT/mTOR and Ras-mitogen-activated protein kinase (MAP kinase) pathway. Activation of these signaling pathways is required for the induction of various activities of IGF-1, including neuroprotection.

**Figure 3 cells-11-00666-f003:**
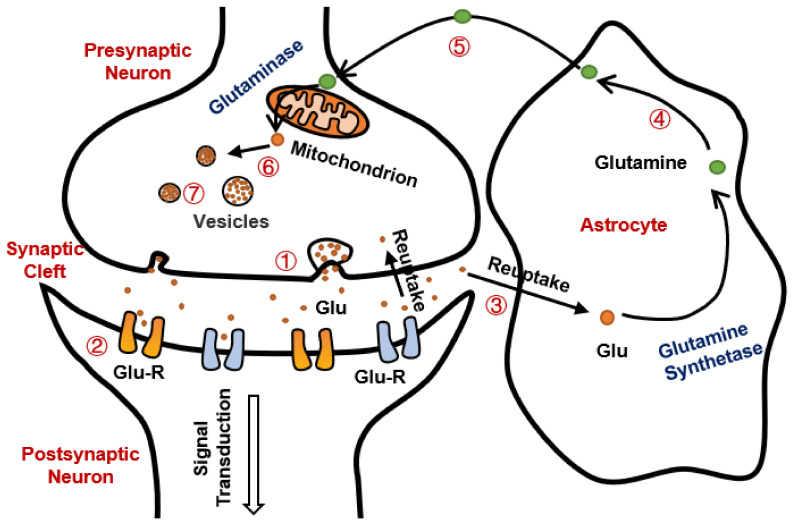
A scheme of the glutamate-glutamine shuttle, focusing on the exchange of glutamate (Glu) and glutamine (Gln) between neurons and astrocytes. ① Glutamate is transported from vesicles to presynaptic membrane and released to synaptic cleft by exocytosis; ② Glutamate in synaptic cleft binds to glutamate receptor (GLU-R); ③ Glutamate undergoes reuptake into astrocytes and neurons; ④ Glutamate in astrocytes is converted to glutamine by the enzyme glutamine synthetase (GS); ⑤ The generated glutamine is transported from astrocytes to neurons by the glutamine transporter; ⑥ Once glutamine enters the neuron, it is converted to glutamate by the mitochondrial enzyme glutaminase; ⑦ Finally, glutamate synthesized from glutamine is concentrated in synaptic vesicles by the vesicular glutamate transporters, thus completing the glutamate-glutamine cycle.

**Figure 4 cells-11-00666-f004:**
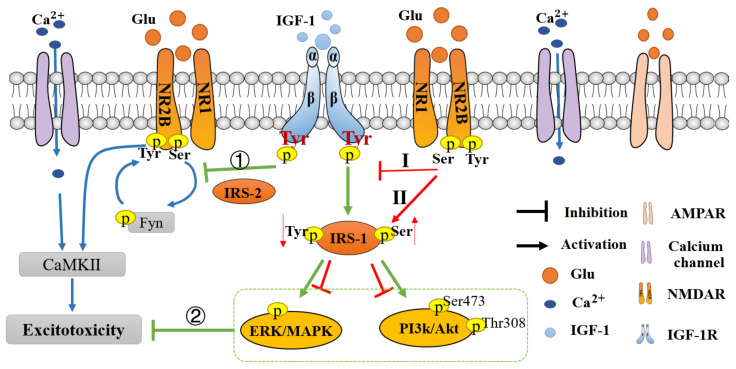
Molecular cross-talk mechanisms between glutamate and IGF-1. The green arrows indicate the mechanism of IGF-1; the red arrows indicate the mechanism of glutamate. IGF-1 inhibits glutamate-induced neurotoxicity by activating the β-subunit of IGF-1R: **①** Inhibiting excitotoxicity induced by glutamate receptor subunit-induced calcium influx, through IRS-2-mediated Ser phosphorylation of NR2B; **②** IRS-1 activated survival downstream MEK/ERK and PI3K/Akt signaling pathways, inhibits excitotoxicity, and confers neuroprotection. However, the overactivation of the NR2B subunit of NMDAR also inhibits the neuroprotective effect of IGF-1: **I.** The activation of the NR2B subunit inhibits the phosphorylation and activation of the β-subunit of IGF-1R, resulting in its uncoupling with IRS-1; **II.** Activation of NR2B subunit enhances IRS-1 serine phosphorylation and inhibits tyrosine phosphorylation, thereby reducing IGF-1R phosphorylation and attenuating IGF-1 survival-promoting effect.

**Figure 5 cells-11-00666-f005:**
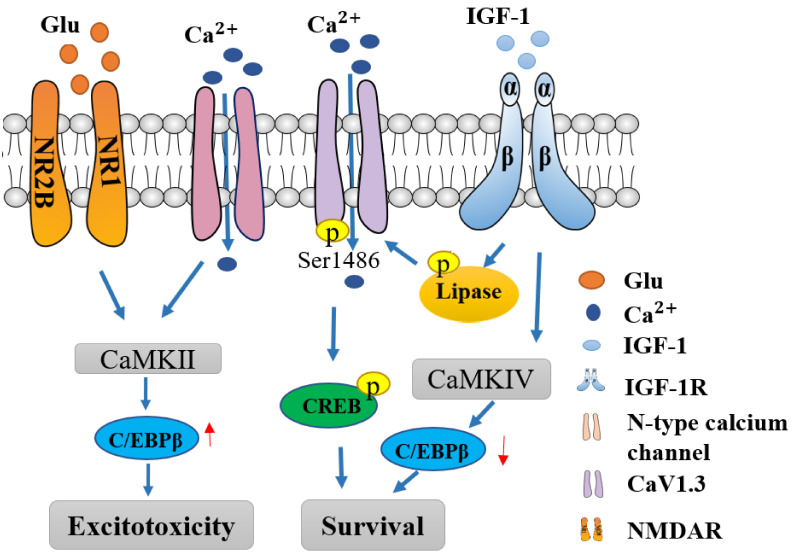
IGF-1 modulation of calcium signaling. Neuroprotective effects of IGF-1 on neurons are achieved by regulation of several subtypes of calcium channels. CaV1.3 channels are modulated by IGF-1 through activation of the phospholipase C and CaMK-II-induced phosphorylation of the CaV1.3 subunit; the phosphorylation of serine residue 1486 of the CaV1.3 subunit rapidly increased CREB levels. In addition, IGF-1 promotes the survival of cerebellar granule neurons by enhancing calcium influx through L-type calcium channels, increased CaMK-IV activity that in turn acts to decrease the CCAAT enhancer nuclear binding proteins (C/EBPβ). Conversely, NMDA receptor-mediated calcium influx rapidly elevates nuclear C/EBPβ and induces excitotoxicity via activation of the calcium-dependent phosphatase, calcineurin.

**Table 1 cells-11-00666-t001:** Preclinical and clinical studies provide evidence that IGF-I confers neuroprotection in animal models and neurological diseases.

Disease	Animal Models	Human Patients	Reference
Ischemic Stroke	Attenuated infarct size with IGF-1 treatment in MCAO and improved post-stroke neurological behaviors.	Inverse correlation between circulating IGF-1 levels and stroke incidence; The levels of IGF-1in the serum is also inversely associated with the neurological deficits following stroke.	[5,111]
Traumatic brain injury (TBI)	IGF-1 is neuroprotective. Functional neurological improvement of motor and cognitive functions in different TBI models.	IGF-1 clinical trials in TBI demonstrate that IGF-1 administration either alone or in combination with GH was safe to humans and successful in improving metabolic parameters in moderate-to-severe TBI patients.	[82]
Amyotrophic Lateral Sclerosis (ALS)	In mouse models of ALS rhIGF-1 delayed disease onset, reduced muscle atrophy, promoted peripheral motor nerve regeneration, and extended life.	Randomized, double-blind, placebo-controlled, phase two and three clinical trials reaffirmed that rhIGF-1 administration was safe and well tolerated in most subjects but efficacy was not statistically significant.	[110]
Alzheimer’s Disease (AD)	In mice with increased cerebral beta-amyloid plaques serum IGF-1 modulated brain levels of beta-amyloid and prevented premature death	Multicenter, cross-sectional study to assess the relationship between IGF-1 and cognitive decline indicated that serum IGF-IGFBP-3 levels were implicated in men with AD. However, a double-blind, multicenter study using growth hormone secretagogue MK-677 which stimulates upregulation and circulation of IGF-1, failed to show efficacy in slowing disease progression.	[109,112,113,114]
Autism spectrum disorder (ASD)- Phelan-McDermid Syndrome (PMS)	I.p. injection of rhIGF-1 in Shank3-deficient mice at clinically approved doses of 0.24 mg/kg/day for 2 weeks reversed the electro-physiological deficits and demonstrated reduced AMPAR-mediated transmission and showed normal LTP comparable to the wild type control mice	A clinical trial using 0.24 mg/kg/day of rhIGF-1 in divided doses, in nine children with PMS (Shank3 deficient) demonstrated safety, tolerability, and efficacy.	[104,115,117]
ASD- Fragile X Syndrome (FXS)	In Fmr1 knockout mice characterized by reduced excitatory synaptic currents, enhanced glutamate receptor dependent-LTD, 100 mg/kg i.p. injection of IGF-1 analog Trofinetide (NNZ-2566) resulted with reduced hyperactivity, improved LSTM and LTP, and normalized social recognition and behaviors.	Phase II randomized, double-blind, placebo-controlled, parallel-group, confirmed the safety, tolerability and efficacy at the high dose of treatment with oral administration of Trofinetide at 35 or 70 mg/kg twice daily, in 72 adolescent or adult males with FXS.	[119,120]
Friedreich’s ataxia (FRDA)	IGF-I in FRDA-like transgenic mice (YG8R mice) conferred neuroprotection and normalized motor coordination.	In a clinical proof of concept pilot study, patients were treated s.c. with IGF-1 therapy with 50 μg/kg twice a day for 12 months and tolerability and decrease in the progression of neurological symptoms was measured, together with long-term stability of cardiac function.	[121,122,123]
Huntington’s disease (HD)	IGF-1 intranasal delivery rescues HD phenotype in YAC128 mice.	In 219 patients with genetically documented HD and in 71 sex- and age-matched controls, IGF-1 serum levels were significantly higher in patients than in controls, indicating somatotropic axis is overactive to confer neuroprotection.	[124,125]
Epilepsy	IGF-I ameliorated hippocampal neurodegeneration and protected against cognitive deficits in an animal model of temporal lobe epilepsy.	57 patients with focal epilepsy and 35 healthy controls were evaluated for IGF-1 level; reduced serum levels of IGF-1were found to correlate with age and cardiovagal function, a parameter of cerebral autoregulation (the breath-hold index). Patients with a longer history of epilepsy, presented higher seizure frequency, and temporal lobe epilepsy and had lower serum levels of IGF-1.	[126,127]

Abbreviations: MCAO, middle cerebral artery occlusion; rhIGF-1, human recombinant IGF-1; GH, growth hormone; i.p., intraperitoneal; s.c., subcutaneous. Fmr1, fragile X mental-retardation protein 1; Shnk3, SH3 and multiple ankyrin repeat domains-3 protein; LTD, long-term depression; LTP, long-term potentiation; LSTM, long short-term memory.

## Data Availability

The datasets generated and/or analyzed, including a large number of TIF images and image analysis data during the current study, are available from the corresponding author on reasonable request.

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
