# Peer review of "Research Progress on Neuroprotection of Insulin-like Growth Factor-1 towards Glutamate-Induced Neurotoxicity"

_cells, 2022, doi:10.3390/cells11040666_

Round 1

Reviewer 1 Report

The reviewed article provides an interesting summary of the topic. I recommend only minor changes.

Introduction
Highlight more the role and importance of IGF-1 in CNS pathophysiology. Clearly specify the topic of the article.

Discussion
The quality of the figures is very poor and should be improved.
What is the clinical significance of the results obtained? This theme needs to be expanded and highlighted more. What is the next step?
The quality of the paper would be greatly enhanced by a figure summarizing the neuroprotective properties of IGF-1. 
Language correction and editorial revision is also required.

Reviewer 2 Report

The English should be revised.

The structures of many sentences are confused, like the one from line 66 to 68, what do the authors want to express? Moreover, some sentences are too long or missed a comma.

Another sentence really confused to my criteria is the one between lines 250-254: “Similarly, in the cochlear afferent synapses contain NRA2  – lacking - AMPARs and NRA2 –containing – AMPARs, the calcium permeable - AMPARs were acutely antagonized with IEM-1460, an antagonist selective for NRA2-lacking AMPARs, to protect from glutamate excitotoxicity, while transmission at NRA2 - containing AMPARs persisted to mediate hearing during the protection.”

There are many redundancies, as in line 149 and 150 the word “adjusting” is repeated in a same sentence, maybe a synonym could be found. Also, in line 150 and 153 the expression “for maintaining” is repeated. Another redundant sentence (lines 236-238): “….in which NRA1/3/4 have high permeability of Ca2+, whereas NRA2 is impermeable to Ca2+. All subunits are permeable to both Na+ and Ca2+ ions with the exception of GluR2, which is uniquely impermeable to Ca2+.” The whole sentence should be simplified.

Please revise the use of Present Progressive (“ing”) form for verbs, as an example in line 175 it seems more appropriate to write: …. glutamate produces…,  instead of glutamate is producing….. There are many examples like this.

Regarding some ideas, it should be explained why the authors postulate that due to the low density or even absence of AMPARs in most areas of the CNS their modulation might reverse early and relevant pathological alterations and demonstrate clinical benefits with much less undesired effects. It is not an idea that derived from that affirmation, on the contrary, if the receptors have a low expression their participation in glutamate excitotoxicity is probably minimum.

In section 4 the references should be updated, there is more recent research that could be cited such as:

DR, Tarantini S et al,  Geroscience. 2019 Apr;41(2):185-208. doi: 10.1007/s11357-019-00065-3. Epub 2019 May 10.

Santi A et al, Cereb Cortex. 2018 Jun 1;28(6):2007-2014. doi: 10.1093/cercor/bhx106

Herrera ML et al, Brain Res Bull. 2021 Oct;175:196-204. doi: 10.1016/j.brainresbull.2021.07.023. Epub 2021 Jul 31

In the first 3 sections, there is much information regarding glutamate receptors, information that is really interesting but seems excessive.

In general, the review is really interested, but need to be rewritten, some parts could be abbreviated, the authors should try to establish a more fluent connection between the ideas.

The figures and the table are really good.
